# A Virtual Instrument for Measuring the Piezoelectric Coefficients of a Thin Disc in Radial Resonant Mode

**DOI:** 10.3390/s21124107

**Published:** 2021-06-15

**Authors:** Francisco Javier Jiménez, Amador M. González, Lorena Pardo, Manuel Vázquez-Rodríguez, Pilar Ochoa, Bernardino González

**Affiliations:** 1Departamento de Electrónica—Física, Ingeniería Eléctrica y Física Aplicada, Escuela Técnica Superior de Ingeniería y Sistemas de Telecomunicación (ETSIST—Campus Sur UPM), Universidad Politécnica de Madrid, 28031 Madrid, Spain; amador.m.gonzalez@upm.es (A.M.G.); pilar.ochoa@upm.es (P.O.); 2Instituto de Ciencia de Materiales de Madrid (ICMM), Consejo Superior de Investigaciones Científicas (CSIC), C/Sor Juana Inés de la Cruz, 3. Cantoblanco, 28049 Madrid, Spain; lpardo@icmm.csic.es; 3Departamento de Ingeniería Telemática y Electrónica, Escuela Técnica Superior de Ingeniería y Sistemas de Telecomunicación (ETSIST—Campus Sur UPM), Universidad Politécnica de Madrid, 28031 Madrid, Spain; m.vazquez@upm.es; 4Escuela Técnica Superior de Ingeniería y Sistemas de Telecomunicación (ETSIST—Campus Sur UPM), Universidad Politécnica de Madrid, 28031 Madrid, Spain; bernardino.gonzalez.fernandez@alumnos.upm.es

**Keywords:** piezoelectric sensors, piezoceramics, *Virtual Instrumentation*, resonators, LabVIEW 2019^®^

## Abstract

In this paper, we describe and present a *Virtual Instrument*, a tool that allows the determination of the electromechanical, dielectric, and elastic coefficients in polarised ferroelectric ceramic discs (piezoceramics) in the linear range, including all of the losses when the piezoceramics are vibrating in radial mode. There is no evidence in the recent scientific literature of any automatic system conceived and implemented as a *Virtual Instrument* based on an iterative algorithm issued as an alternative to solve the limitations of the ANSI IEEE 176 standard for the characterisation of piezoelectric coefficients of thin discs in resonant mode. The characterisation of these coefficients is needed for the design of ultrasonic sensors and generators. In 1995, two of the authors of this work, together with other authors, published an iterative procedure that allowed for the automatic determination of the complex constants for lossy piezoelectric materials in radial mode. As described in this work, the procedures involved in using a *Virtual Instrument* have been improved: the response time for the characterisation of a piezoelectric sample is shorter (approximately 5 s); the accuracy in measurement and, therefore, in the estimates of the coefficients has been increased; the calculation speed has been increased; an intuitive, simple, and friendly user interface has been designed, and tools have been provided for exporting and inspecting the measured and processed data. No *Virtual Instrument* has been found in the recent scientific literature that has improved on the iterative procedure designed in 1995. This *Virtual Instrument* is based on the measurement of a unique magnitude, the electrical admittance (*Y = G + iB*) in the frequency range of interest. After measuring the electrical admittance, estimates of the set of piezoelectric coefficients of the device are obtained. The programming language used in the construction of the *Virtual Instrument* is LabVIEW 2019^®^.

## 1. Introduction

Piezoceramics present the so-called piezoelectric effect, which consists of the polarisation of the material following a certain polar axis when the piezoceramic is subjected to mechanical traction or compression [1,2]. The polarisation of the material causes an accumulation of a charge in one direction or another of its polar axis, manifesting as a potential difference in the faces of the crystals [3,4,5,6,7,8]. This phenomenon is called the direct piezoelectric effect. Piezoceramics also present the inverse piezoelectric effect: if the piezoceramic is subjected to an external electric field, it will be compressed or stretched depending on the value and direction of the electric field [1]. Figure 1 shows a depiction of the piezoelectric effect.

Piezoceramics are used as sensors and/or generators, and their applications are constantly expanding [9,10,11,12,13], especially in industrial applications and in a wide range of electronic devices. The electromechanical characterisation of these devices must be as reliable and accurate as possible such that the electromechanical behaviour of the sensor or actuator is perfectly described [14,15].

The most extended characterisation method of these types of piezoceramic materials is well-known as the resonance method. It is based on the inverse piezoelectric effect (Figure 1b) that provokes a vibration when an AC electric field in a certain interval of frequencies is applied to the material. This method is described in the ANSI IEEE 176 [2] standard, which provides a procedure to estimate the electromechanical, dielectric, and elastic coefficients in resonant modes from measurements of the impedance/admittance of the sample, assuming low energy losses. Therefore, the values of the coefficients calculated by the ANSI IEEE 176 method are real numerical values (i.e., without an imaginary part). Numerous papers have been published on the characterisation of piezoelectric ceramics based on this standard [16,17]; however, characterisation of the dielectric and elastic coefficients, taking into account material losses, has provided further research motivation for materials scientists and engineers [18,19,20,21,22,23,24,25,26,27,28,29,30,31], which have used both the resonance method and other direct measurements.

In previous works, the authors of this paper exposed some difficulties presented by the ANSI IEEE 176 standard in terms of calculating exact material coefficients when the energy losses are not ignored. For this reason, they proposed an iterative, programmable method, based on Smits’s [22] method, that allows for estimation of the coefficients while taking into account the energy losses in the material by providing an estimate of the coefficients with complex numerical values (i.e., with real and imaginary parts) [18,32,33], as the imaginary part of the coefficients represents the non-negligible losses in the material.

The iterative method consists of searching for resonance frequencies in samples with different geometries (see Figure 2) and measuring the electrical admittance (*Y = G + iB*) in the frequency range of interest in the laboratory using an impedance analyser. With the expressions of the real and imaginary parts of the electrical admittance, Y, a system of nonlinear equations can be obtained. The resolution of this system of nonlinear equations is solved using an iterative numerical method based on the measurements obtained. The constructed *Virtual Instrument* consists of hardware for excitation and measurement in the sample as well as software for control and management of the measured data. No *Virtual Instrument* that performs this type of automatic characterisation of these devices has been found in the recent scientific literature. The authors of this paper have modified the iterative procedure [18] for better accuracy in estimating the coefficients. There is also no evidence in the recent scientific literature of any *Virtual Instrument* based on an iterative procedure based on a single measure of admittance as a function of frequency.

To carry out estimation of the coefficients using the iterative numerical method, different versions of measured systems have been developed using different programming languages, including QUICKBASIC^®^ (1994), JAVA^®^ (2003), and LabVIEW 2008^®^. In all previous versions, the number of samples acquired and the iterative method were the same.

The versions mentioned are composed of a set of programs that are independent of each other and that implement the estimation of piezoelectric coefficients under different modes of vibration, as shown in Figure 2. Figure 2 includes a non-standard, thickness poled and longitudinally excited shear plate (Figure 2b), amenable to achieve the monomodal resonance needed for the application of the resonance method [34].

The original versions lack the capabilities provided by modern programming languages, such as libraries of mathematical analysis functions, functionality for the presentation of data, ease of installation of software on different platforms, and ease of exporting data, among others.

Our motivation for building a *Virtual Instrument*, the tool described in this paper, was to improve upon previous versions, specifically in the following aspects:Improve the automatic procedure for the calculation of piezoelectric coefficients in resonant mode.Increase the speed of processing and the calculation of constants.Increase the accuracy of the results.Improve the user interface by making it friendly and intuitive.Provide the system with tools for the exploration and export of the acquired and processed data.Make the *Virtual Instrument* software easily readable, expandable, and scalable.

Figure 2 shows the five resonant modes of interest for determination of material coefficients. We chose to construct a *Virtual Instrument* so that estimation is calculated with respect to the radial resonant mode of a thin disc as it is the most complicated procedure to implement and because there are many electronic devices that use this type of geometry and vibration mode.

The software architecture of the *Virtual Instrument* designed and developed allows the instrument to be expanded and scaled so that the inclusion of the analogue iterative procedures for the resonant modes’ vibration shown in Figure 2 would not involve interaction with the code already implemented for the thin disc in radial resonant mode.

In addition, LabVIEW 2019^®^ can perform operations with extended length floating-point numbers of 128 bits (real and imaginary), and it allows substantially to improve the speed and processing capacity of the designed tool.

## 2. *Virtual Instrument* Description

The system conceived for a *Virtual Instrument* consists of a hardware part and a software part running on a computerised system (PC). A block diagram of the system is shown in Figure 3.

The computerised system of a *Virtual Instrument* includes data acquisition hardware, drivers for the peripherals, and a software application to perform the functions of the measuring instrument [35].

*Virtual Instrument*s, more generally, represent a significant shift from traditional instrumentation systems, which are typically focused on one measurement instrument towards computer systems that can improve productivity, visualisation, digital signal processing, and connectivity capabilities.

A *Virtual Instrument* adheres to the slogan “software is the instrument” [36,37]. It is the software, using the appropriate hardware, that defines the instrument’s functions.

The input data of a *Virtual Instrument* are:The measured admittance of the sample, which is placed in a 16065A Ext Voltage Bias Fixture HP sample holder connected to an HP4194A impedance analyser. The sample holder has to meet the two conditions of allowing the sample to vibrate freely, by not using rigid terminals that will clamp the sample, and avoiding the addition of admittance components that could modify the measurement of the complex admittance of the sample. Figure 4 shows how the sample is connected:The start frequency (Hz), that is, the initial excitation frequency.The stop frequency (Hz), that is, the final excitation frequency.The sample radius *a* (mm) and sample thickness *t* (mm).

The data calculated [2,18] and reported by a *Virtual Instrument* are:
$c_{11}^{p}$, the elastic stiffness coefficient (N/m^2^) in the radial plane:(1)$$c_{11}^{p}=-\frac{s_{11}^{E}}{{\left(s_{11}^{E}\right)}^{2}-{\left(s_{12}^{E}\right)}^{2}\;}$$
where $s_{pq}^{E}$ is the compliance, which is the relationship between the relative deformation in the *p-*axis and the mechanical stress in the *q* direction for an electric field *E*.$\sigma^{p}$, the planar radial Poisson’s ratio:(2)$$\sigma^{p}=-\left(\frac{s_{12}^{E}}{s_{11}^{E}}\right).$$$\varepsilon_{33}^{T}$, the relative dielectric permittivity when an electric field is applied in the direction of polarisation (i.e., direction 3) at a constant mechanical stress (T = 0, “free” permittivity). Therefore, $\varepsilon_{33}^{T}=\epsilon_{0}\cdot \epsilon_{33}^{T}$, where $\epsilon_{0}=8.8541878176\cdot 10^{-12}{\;C\;}^{2}/\left(N\cdot m^{2}\right)\;$.*d*_31_, the proportion between the dimensional variation (Δr) of the piezoelectric material (in meters) and the difference of potential applied in volts in axis 3, as well as between the generation of electric charges and the force applied in the material.*k_p_*, planar radial piezoelectric coupling:(3)$$k_{p}^{2}=\frac{2\cdot d_{31}^{2}}{\epsilon_{0}\cdot \epsilon_{33}^{T}\cdot \left(s_{11}^{E}+s_{12}^{E}\;\right)}.$$Measurement graphs: The *Virtual Instrument* provides several graphs from the measured data of electrical admittance, *Y = G + iB*, as well as utilities for data exploration.Logged data file: The *Virtual Instrument* records the measured and calculated data in files that can be exported to other applications. Moreover, all the intermediate steps of the calculations performed in the iterations are recorded.

The electrical admittance of the disc at the radial resonance frequency follows Equation (4), which is Equation (117) of the “IEEE Standard of Piezoelectricity” [2]:(4)$$Y=\frac{i\cdot 2\pi f\cdot \varepsilon_{33}^{T}\cdot \pi a^{2}}{t}\left[1-k_{p}^{2}\right]\left(\frac{{ℑ}_{1}\left(\eta \right)-1+\frac{\sigma^{p}+k_{p}^{2}}{1-k_{p}^{2}}}{{ℑ}_{1}\left(\eta \right)-1+\sigma^{p}}\right)$$
where ${ℑ}_{1}\left(\eta \right)$ is a complex variable function called Onoe’s function [18], which follows the expression:(5)$${ℑ}_{1}\left(\eta \right)=\frac{\eta \cdot J_{0}\left(\eta \right)}{J_{1}\left(\eta \right)}$$
where $J_{0}$ and $J_{1}$ are the zeroth- and first-order first-class Bessel functions, respectively, with argument η as follows:(6)$$\eta =\frac{2\pi f\cdot a}{\sqrt{\frac{\rho }{c_{11}^{p}}}}$$

With all of the above, along with the information of the conducted measurements, the iterative numerical method is applied, which calculates the coefficients $\varepsilon_{33}^{T}$, $\sigma^{p}$, $c_{11}^{p}\;$, *d*_31_, and *k_p_* while taking into account the energy losses. The numerical iterative method has been described in detail in [18].

### 2.1. Virtual Instrument Measurements

The measurements required by the iterative numerical method consist of searching for the singular frequencies of *Y = G + iB* and the electrical impedance *Z = R + iX* within a certain frequency range. This frequency range must cover at least one overtone of the mode of interest.

The measurements that a *Virtual Instrument* must obtain, in order to start the iterative method [18] from the data measured with the impedance analyser (HP4194A), are as follows:*f*_1s_ and *f_*2*s_*, the frequencies where *G* is maximal in the fundamental tone and in the overtone, respectively. The values of these frequencies are used as initial values in the iterative method.*f_p_*, the frequency where *R* is maximal.Δ*f_s_ =* |*f_Bmax_ − f_Bmin_*|, where *f_Bmax_* and *f_Bmin_* are the frequencies at which B becomes maximal and minimal, respectively, in the fundamental tone. The values of these frequencies are used as initial values in the iterative method.

Therefore, a *Virtual Instrument* must measure *Y* within a frequency range, calculate Z from the measured data, and search for the frequencies of interest with an adequate resolution. It is at this point that the design of a *Virtual Instrument* presents important improvements with respect to that presented in [18], and to the previous software versions mentioned in the Introduction.

The available impedance analyser (HP4194A) was limited to 401 measurement points between the initial sweep frequency, *Start frequency* (Hz), and the final sweep frequency, *Stop frequency* (Hz).

Typically, the frequency ranges to be swept on the discs under a study of 10–15 mm radius and approximately 1 mm thickness is around 200 kHz. This means that, if the frequency search were done in a single step, the frequency resolution of the measurement would be very low, which worsens the accuracy of estimates of the coefficients after applying the iterative method. This forces the operator of the previous software to carry on a number of measurements to ensure the precision of this frequency determination.

To solve this problem, all the process has been automated in the *Virtual Instrument*. We have chosen to implement a solution where the user determines the frequency range in which the frequencies of interest are located. Then, automatically, the VI performs three contiguous frequency sweeps of 400 points (1200 points), and the search for the values of *f*_1s_, *f_*2*s_*, *f_p_*, *f_Bmax_*, and *f_Bmin_.* These values are the initial values to start the implemented iterative method. To do so, there is a previous step of calculation of real and imaginary parts of the measured complex admittance (*Y* = *G* + i*B*) and, also, of its inverse, the complex impedance (*Z = R* + i*X*).

Figure 5 presents the procedure.

### 2.2. Virtual Instrument Software

The system software was written using the graphical programming language LabVIEW 2019^®^, which is the most widely used language for *Virtual Instrument*ation. The programs developed with LabVIEW 2019^®^ consist of two main windows, the front panel (user interface), and the block diagram (program code). The programming is modular, where each module is built with one or more files called SubVIs [38].

In this work, the *Virtual Instrument* was designed using a state machine architecture for measurement and control. This design pattern is very suitable for the design of complex measurement equipment in which different functionalities are required, namely, the admittance measurements and calculations for determination of coefficients in the four modes specified in Figure 2. A state is a situation in which the machine can be found. Usually, in a given state, actions or operations associated with this state are performed; furthermore, there will be a decision code to decide what state the machine should evolve to. A machine consisting of states is graphically represented as a diagram of states. The programming language LabVIEW 2019^®^ allows the user to implement a state machine with a simple design pattern, which allows for the development of complex programs, making VI scalable, maintainable, and readable. Figure 6 shows the used LabVIEW 2019^®^ template.

Figure 7 shows the state diagram of the designed *Virtual Instrument* software. The state diagram is intended to be self-explanatory for readers with knowledge in the software. The transition between the states is caused by the events produced when the user interacts with the front panel controls.

If the STOP control is pressed, the program will stop. In the CALCULATION state, the numerical iterative method is implemented, which provides the coefficients. Figure 8 shows a flowchart of the iterative process, which has been explained in detail in [18]. As can be seen, the iterative calculation can begin with the frequencies described above (*f*_1s_, *f_*2*s_, f_Bmax_,* and *f_Bmin_*).

With these measurements, we obtain *k_p,initial_*, *f_*1*,initial_*, *f_*2*,initial_*, *η_*1*_*, and *η_*2*_*, where:(7)$$k_{p,initial}=1-\frac{f_{1s}}{f_{p}}\;,$$
(8)$$f_{1,initial}=\frac{f_{1s}}{\sqrt{1+\left|k_{p,initial}\right|}}\;,\;\;\;\;\;\;\;\;\;\;\;\;\;\;$$
(9)$$f_{2,initial}=\frac{f_{1s}}{\sqrt{1+\left|k_{p,initial}\right|}}\;,\;$$

The initial guess value of σ**^p^** (σ_p__, initial_) and $\eta_{1}$ are obtained by solving the system of equations of the Equation (10). The system is solved by the iterative method as described in [18]:(10)$$1-\sigma^{p}-{ℑ}_{1}\left(\eta_{1}\right)=0\;\;\;1-\sigma^{p}-{ℑ}_{1}\left(\eta_{2}\right)=0\;\;\;\eta_{2}=\eta_{1}\cdot \frac{f_{2s}}{f_{1s}}$$

The initial values of σ**^p^** $\eta_{1}$, and $\Delta f_{s}=\left|f_{Bmax}-f_{Bmin}\right|$ allows to calculate the initial value of $c_{11,\;initial}^{p}$ as described in [8].
(11)$${\left(c_{11}^{p}\right)}^{\prime }={\left(\frac{2\pi f_{1s}\cdot a}{\eta_{1}}\right)}^{2}\cdot \rho \;\;\;\;\;\;\;\;\;\;\;real\;part\;\;\;\;\;$$
(12)$${\left(c_{11}^{p}\right)}^{\prime \prime }=-{\left(\frac{\Delta f_{s}}{f_{1s}}\right)}^{2}\cdot {\left(c_{11}^{p}\right)}^{\prime }\;\;\;\;\;\;\;imaginary\;part$$

From here, the state machine continues the calculations following the iterative method shown in Figure 8.

## 3. Experimental Results and Discussion

The following shows and describes the practical realisation of the characterisation of a thin piezoelectric disc using a *Virtual Instrument*. For this purpose, a commercial disc of soft lead titanate zirconate, with reference PZ27 of Meggit Ferroperm^TM^ Piezoceramics [39], with a density *ρ* = 7700 kg/m^3^, radius = 15 mm, and thickness = 1 mm. Figure 9 shows the front panel of a *Virtual Instrument*.

Where:
C_11^p ≡ $c_{11}^{p}$: stiffness coefficient (N/m^2^)O^p ≡ σ^p^: Planar Poisson’s ratiod_31 ≡ *d*_31_: Charge piezoelectric constant (C/N or m/V)E_33^T ≡ $\epsilon_{33}^{T}$: Dielectric permittivity (relative)Kp (%) ≡ *k_p_*: Planar radial piezoelectric coupling

The *Sweep Frequency* was performed between 100 Hz and 0.9 MHz, as indicated in Figure 5. The data obtained in the first sweep are shown in Figure 10.

Figure 11, Figure 12 and Figure 13 show the data obtained from a PZT after performing the Automatic Search step, indicated in the flowchart in Figure 5. The data can be viewed by selecting the corresponding tabs in the *Virtual Instrument* front panel.

With the automatically found values of *f_*1*s_*, *f_*2*s_*, *f_p_*, *f_Bmax_*, and *f_Bmin_*, the iterative method described in Figure 8 begins. The iterative method is executed in the CALCULATION state (see Figure 7).

In addition, the *Virtual Instrument* has, as a novelty, the added functionality which allows to save all the intermediate steps of the calculations performed in the iterative procedure. The intermediate steps are shown in the “Information” section of the *Virtual Instrument* front panel (see Figure 14).

### 3.1. Processing/Computation Time

To quantify the time taken by the *Virtual Instrument* to characterise the tested sample shown in this paper, we used the data acquisition time estimate from the manufacturer of the HP4194A [40] as well as the LabVIEW 2019^®^ *Profile Performance* tool [41]. The computation time estimate in this section is the sum of the execution time of the states “MEASUREMENTS”, “GRAPH”, and “CALCULATION”.

We considered the maximum time between samples in the HP4194A manufacturer’s specifications for the execution time of the state “MEASUREMENTS”.

Ten runs were carried out to estimate the processing time. Results are shown in Table 1.

Therefore, it is possible to estimate the average time that the *Virtual Instrument* took to characterise the sample:
T_characterisation_ (ms) = 4440 + 7.8 + 15.9 × 6 + 2.1 = 4540.5.

### 3.2. Comparation with JAVA (2003) Software

A comparison was made between the obtained results and those obtained using JAVA software (i.e., using the characterisation program constructed in JAVA in 2003).

Table 2 shows the results of the implemented *Virtual Instrument* and the JAVA version (2003).

The differences in the calculation procedures leading to the values shown in Table 2. were due to the improvements made to the implemented *Virtual Instrument*, which are summarised below:In the *Virtual Instrument* program, the value of $c_{11}^{p}$ is accepted when the relative value of the difference of two consecutive values of $c_{11}^{p}$ is lower than $10^{-8}$ GN/m^2^ while in the JAVA (2003) program, it is accepted when it is lower than $10^{-6}$ GN/m^2^.The values of *f_*1*_* and *f_*2*_* are accepted in *Virtual Instrument* when the relative value of the difference of two consecutive values of f_2_ is lower than 5·10^−4^ Hz while in JAVA (2003), they are accepted when this is lower than 10^−3^ Hz. For this reason, the values of $\varepsilon_{33}^{T},\;d_{31},\;\sigma^{p},\;\mathrm{and}\;k_{p}$ were accurately estimated.The number of points acquired in the search for the initial values of the numerical method is three times greater. This gives place to high accuracy of calculation for all values.The frequencies of interest, *f*_*1*s_, *f_*2*s_*, *f_p_*, *f_Bmax_*, and *f_Bmin_*, are automatically searched for with an adequate resolution. This improved the accuracy of calculation for all values.The calculation and results’ presentation time was much shorter when using the version of *Virtual Instrument* implemented in this work.

Thanks to these differences, the iterative method can be used with all the benefits of the up-to-date programming advantages of the *Virtual Instrument* and, as data in Table 2 shows, with great accuracy, lower processing time and very good matching with the results of previous software.

## 4. Conclusions

The proposed *Virtual Instrument* improved the procedures of previous versions for the automatic calculation of piezoelectric coefficients from admittance measurements at the radial extensional resonant mode of thickness poled discs.

The software architecture used, in addition to the floating-point resolution of real and complex numbers, substantially improved the speed and processing capacity of the designed tool.

The results in Table 2 and the discussion in Section 3 reveal that, together with highly automatized measurements and all the advantages related to the *Virtual Instrument*, highly accurate results of the calculation that matches the results from previous software were achieved.

The *Virtual Instrument* has a simple and friendly user interface. The graphs shown allow for visualisation of the admittance in polar form (Y–theta) or in Cartesian form (*Y = G + iB*), and of the impedance in Cartesian form (*Z = R + iX*) as a function of the frequency. These graphs are also equipped with advanced visualisation tools (e.g., focus, zoom, cursors, etc.).

The *Virtual Instrument*, as implemented in this work, records the measured and calculated data in exportable files (ASCII) for use in other applications. In addition, the *Virtual Instrument* has, as a novelty, the added functionality which allows all the intermediate steps of the calculations performed in the iterative procedure to be recorded.

The *Virtual Instrument* has the advantage of making use of the extensively tested iterative method of Alemany et al. [32] which allows characterizing even samples with very weak piezoelectric coefficients, as well as samples with strong losses, something that the method proposed in ANSI IEEE 176 does not allow.

The software architecture used in the *Virtual Instrument* makes it easily readable, expandable, and scalable. A future line to be developed is to extend the *Virtual Instrument* to include the implementation of the resonance modes shown in the geometries of Figure 2. In this way, a complete characterisation of the elastic-piezo-dielectric matrix of a material in the geometries typically used in piezoceramic devices could be possible.

## Figures and Tables

**Figure 1 sensors-21-04107-f001:**
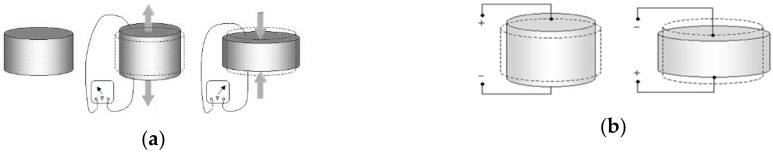
Piezoelectric effect: (**a**) Direct effect; (**b**) Inverse effect.

**Figure 2 sensors-21-04107-f002:**
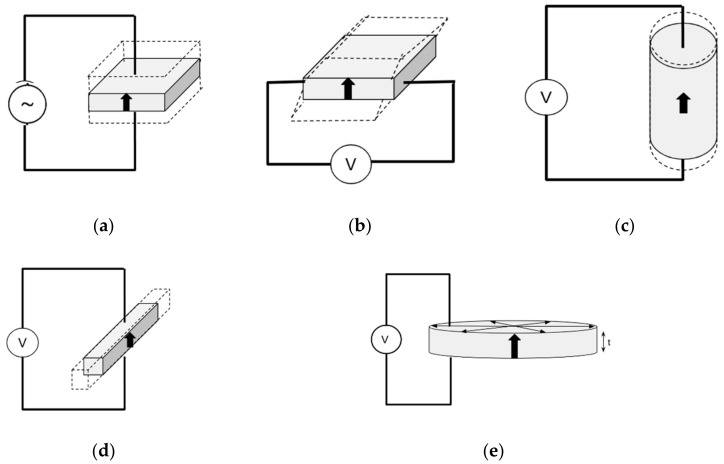
Schematic of the resonant modes of vibration of ceramic resonators: (**a**) Thickness extensional of a thickness poled thin plate; (**b**) Shear of a non-standard thickness poled thin plate; (**c**) Length extensional of a longitudinally poled cylinder or bar; (**d**) Length extensional of a thickness poled thin bar; (**e**) Radial extensional of a thickness poled thin disc.

**Figure 3 sensors-21-04107-f003:**
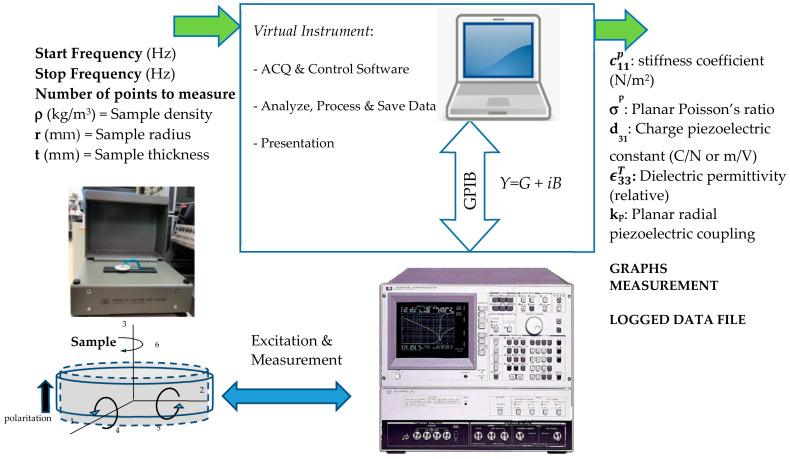
Block diagram of the proposed measurement system.

**Figure 4 sensors-21-04107-f004:**
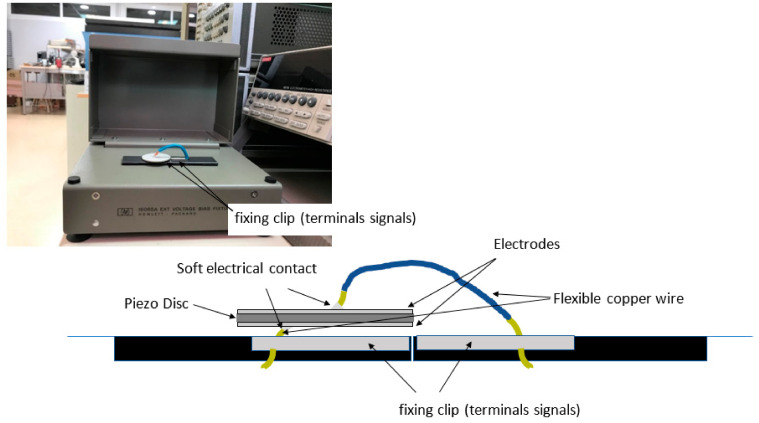
Image of experimental setup and schematic of sample.

**Figure 5 sensors-21-04107-f005:**
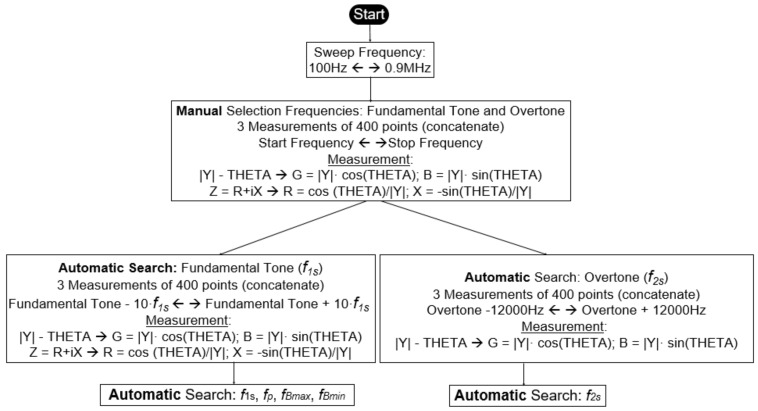
Procedure for searching for the frequencies *f*_1s_, *f_p_*, *f_Bmax_*, and *f_Bmin_* from the fundamental tone and *f_2s_* from the first overtone.

**Figure 6 sensors-21-04107-f006:**
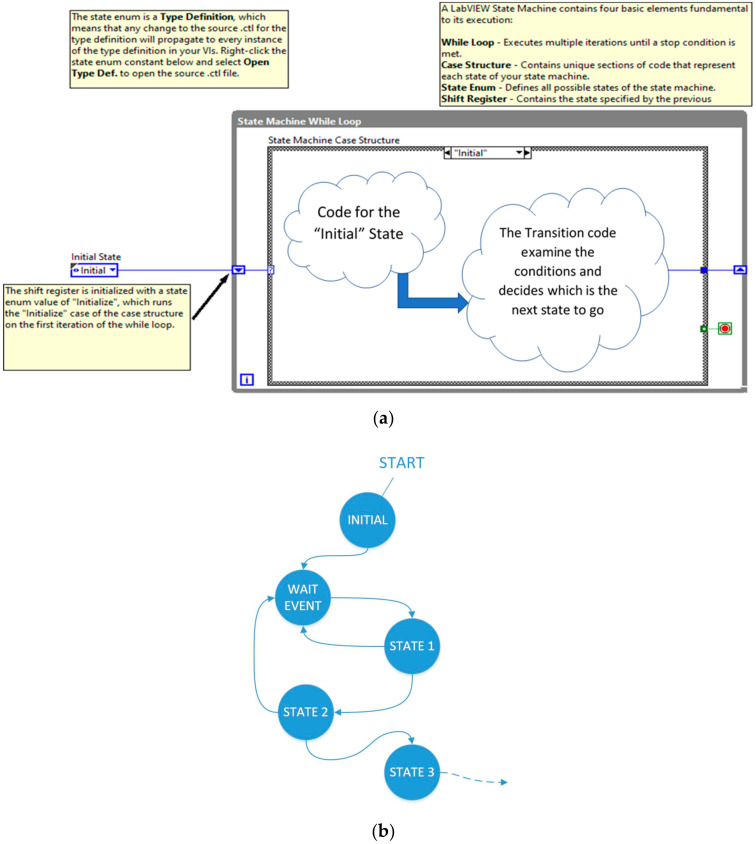
(**a**) State machine LabVIEW 2019^®^ template; (**b**) Typical state diagram.

**Figure 7 sensors-21-04107-f007:**
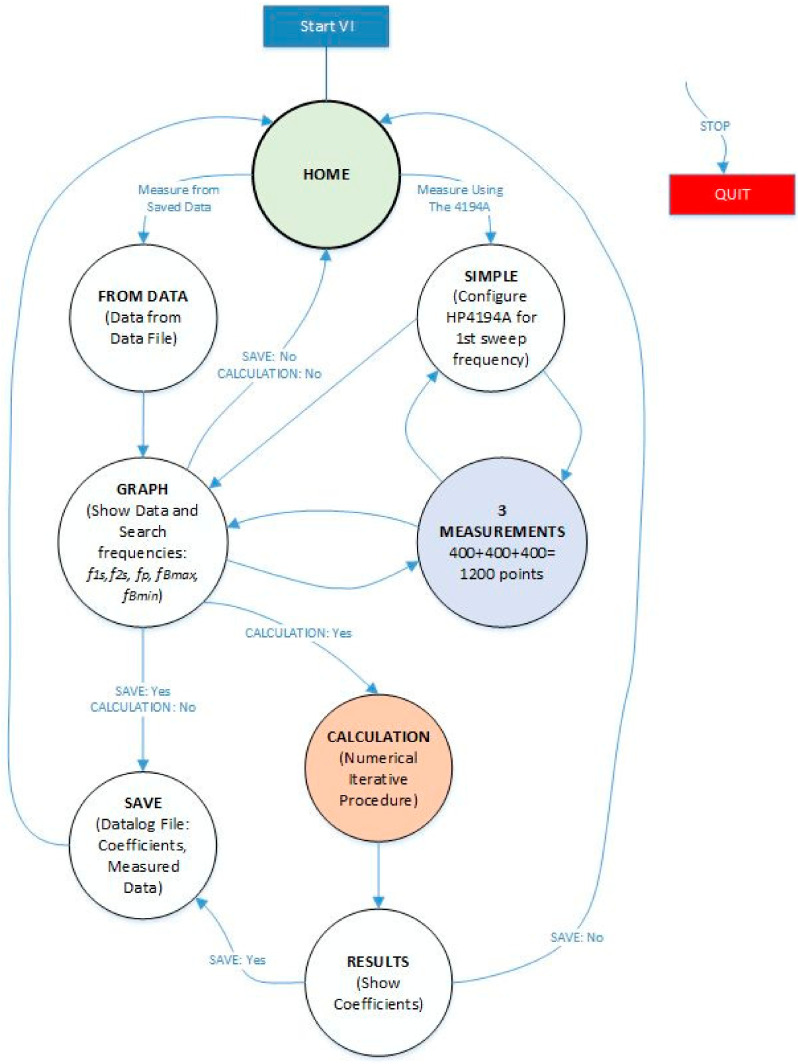
State diagram of the *Virtual Instrument* software.

**Figure 8 sensors-21-04107-f008:**
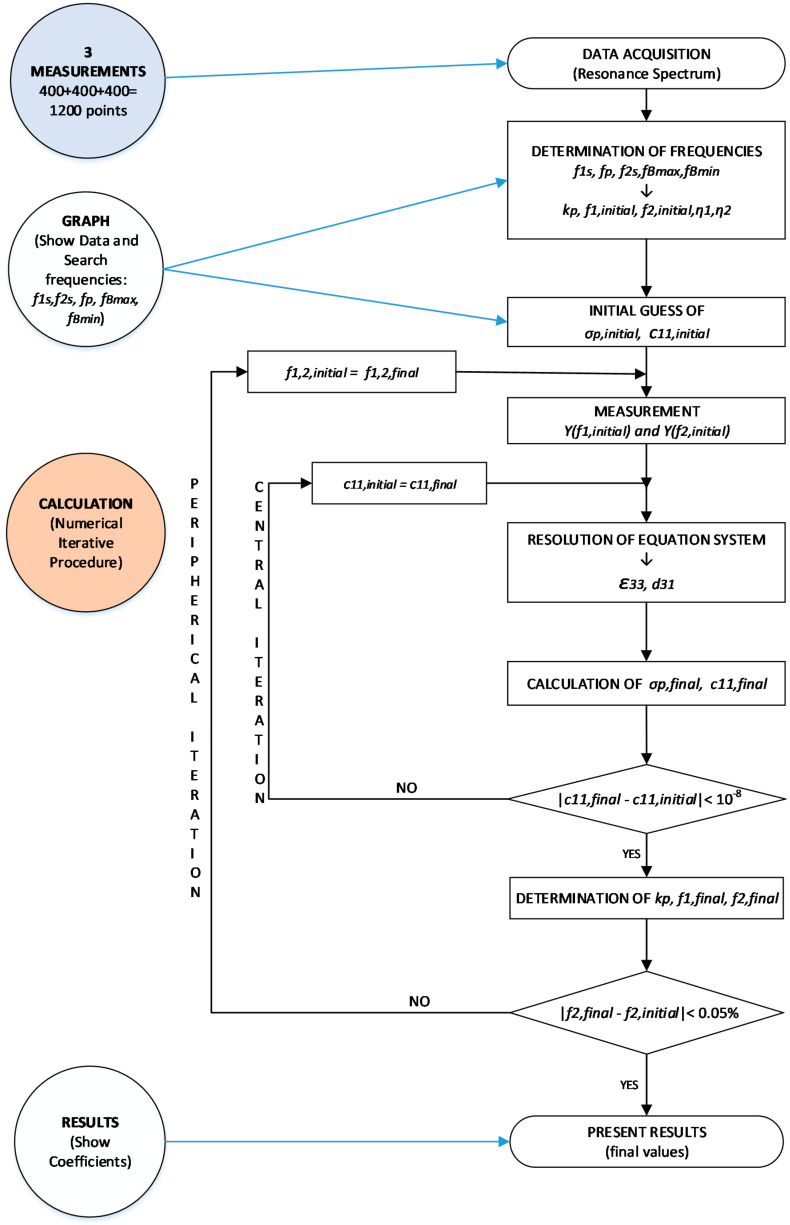
Flowchart of the iterative method.

**Figure 9 sensors-21-04107-f009:**
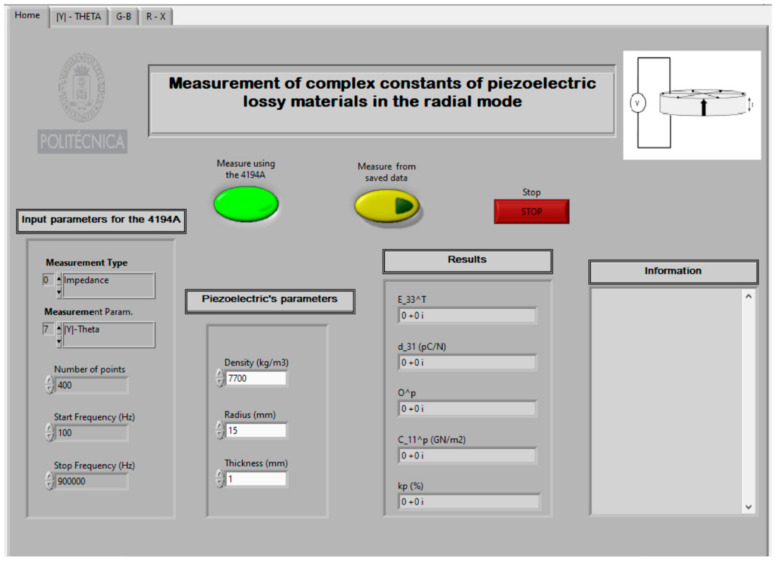
*Virtual Instrument* front panel.

**Figure 10 sensors-21-04107-f010:**
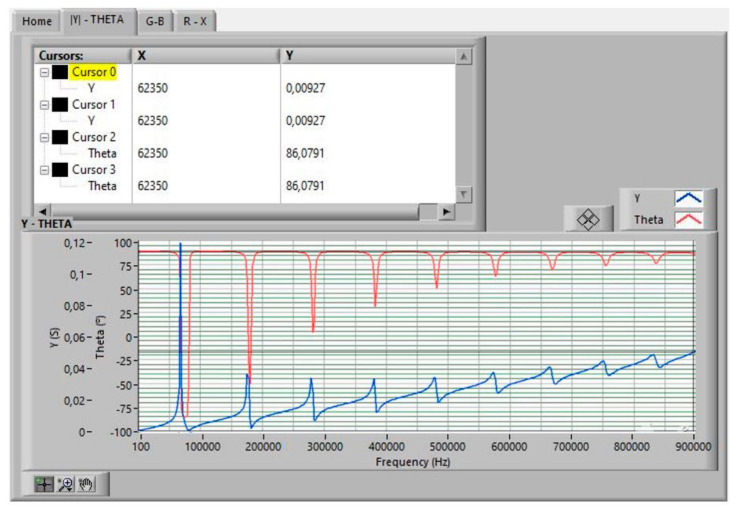
Sweep frequency data measurement results.

**Figure 11 sensors-21-04107-f011:**
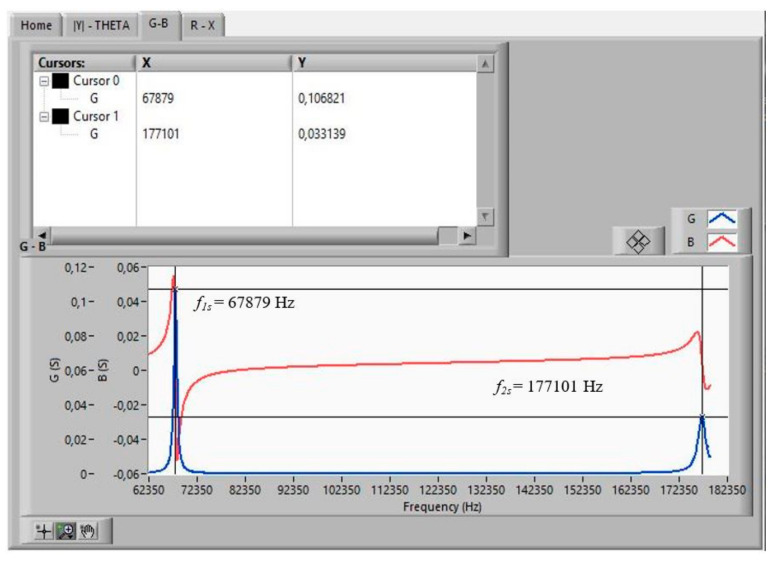
Searched frequencies after the automatic search step: *f_1s_*, *f_2s_*.

**Figure 12 sensors-21-04107-f012:**
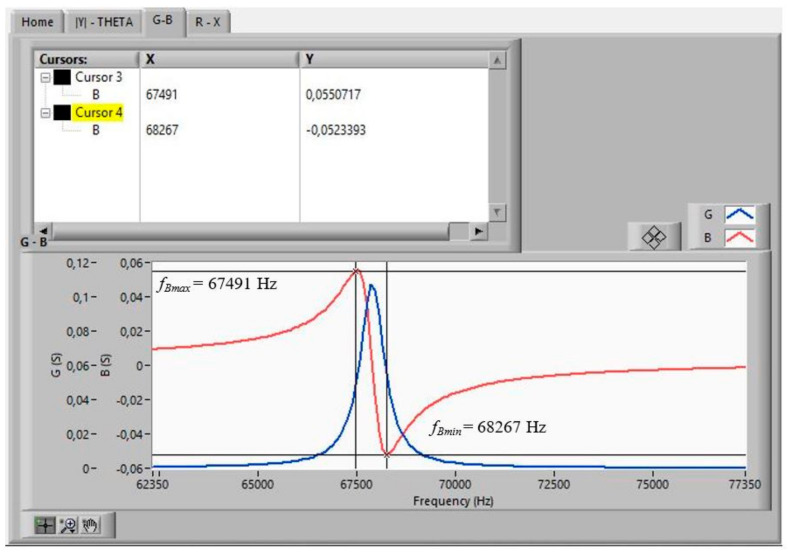
Searched frequencies after the automatic search step: *f_Bmax_*, *f_Bmin_*_._

**Figure 13 sensors-21-04107-f013:**
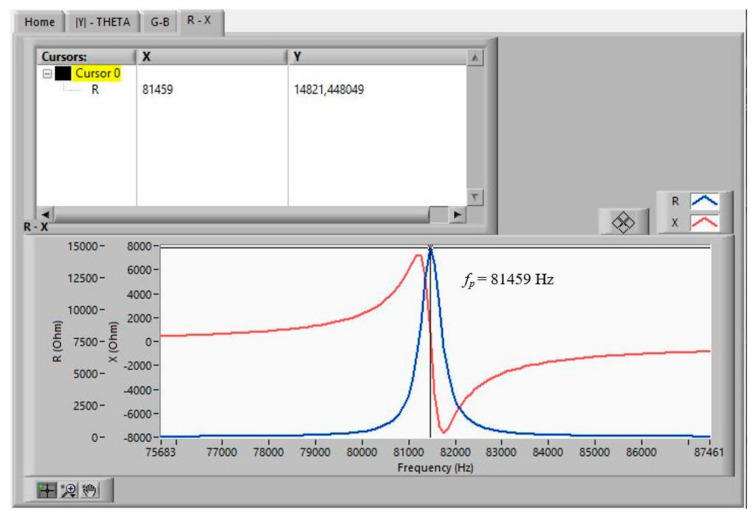
Searched frequencies after the automatic search step: *f_p_*_._

**Figure 14 sensors-21-04107-f014:**
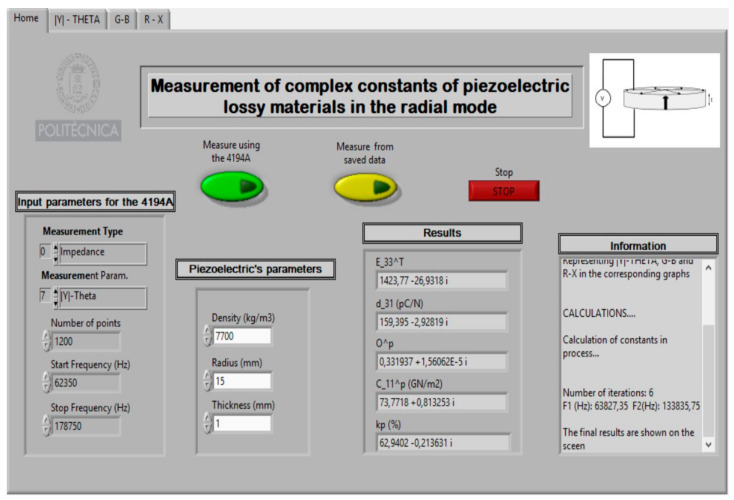
Results of test.

**Table 1 sensors-21-04107-t001:** Profile performance.

State	#Runs	Average (ms)	Shortest (ms)	Longest (ms)
MEASUREMENTS	400 points 3 measurements 3.7 ms = 4440 ms			
GRAPH	10	7.8	7.1	8.4
CALCULATION (6 iterations)	10	15.9 × 6	15.1 × 6	17.0 × 6
RESULTS	10	2.1	1.3	2.6

**Table 2 sensors-21-04107-t002:** Results of the calculation with the JAVA version and the implemented Virtual Instrument.

	JAVA (2003)	*Virtual Instrument*
$c_{11}^{p}\left(GN/m^{2}\right)$	73.837502 + 0.80914118 i	73.7718 + 0.813253 i
$\sigma^{p}$	0.33042568 + 3.4927518·10^−5^ i	0.331937 + 1.56062·10^−5^ i
$\varepsilon_{33}^{T}$	1421.2551 − 26.966124 i	1423.77 − 26.9318 i
$d_{31}\left(pC/N\right)$	159.00174 − 2.919189 i	159.395 − 2.92819 i
$k_{p}\left(\%\right)$	62.632594 − 0.21235708 i	62.94022 − 0.213614 i
$f_{1}\left(Hz\right)$	63,848.44	63,827.35
$f_{2}\left(Hz\right)$	133,641.36	133,835.75
T_characterisation_ (ms)	≈8000	≈5000

## Data Availability

The data presented in this study are available on request from the corresponding author. The data are not publicly available due to privacy.

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
