# Peer review of "A Virtual Instrument for Measuring the Piezoelectric Coefficients of a Thin Disc in Radial Resonant Mode"

_sensors, 2021, doi:10.3390/s21124107_

Round 1
Reviewer 1 Report
Authors proposed interesting approach of graphic interface for piezoelectric coefficients of a ultrasonic transducers. As far as I know, the piezoelectric coefficients are measured by acoustic measurement systems of Onda Corp.
Authors showed cost-effective solutions to measure the piezoelectric coefficients with lossy medium in a radial resonant mode. The experimental data with electrical impedance with several parameters are reasonably good to be seen.
Overall, the manuscript is well written. However, some Figure qualities are not very clear to be seen. Some references should be added. Therefore, the manuscript could be minor revision if authors follow the suggestions.
1. Please increase label sizes in Figures 5 and 6.
2. In Figure 8, please increase the comment labels.
3. Please increase the font sizes of Figure 11.
4. In Figure 13, fonts are not very clear.
5. Please provide the data availability and acknowledgement sections.
6. Please provide access date of "Available Online" in the reference.
7. Please check reference format of MDPI.
8.Please provide the reference of the sentence (The polarisation of the material causes an accumulation of charge in one direction or another of its polar axis, manifesting as a potential difference in the faces of the crystals) with the reference (Kim, K., & Choi, H. (2021). High-efficiency high-voltage class F amplifier for high-frequency wireless ultrasound systems. PloS one, 16(3), e0249034. ).
9.Please provide the reference of the sentence (For this purpose, a PZT27 sample was chosen, with a density~ ) with the reference (Maksymovych, Peter, et al. "Tunable metallic conductance in ferroelectric nanodomains." Nano letters 12.1 (2012): 209-213. ) or another reference.
10.Please provide the reference of the sentence ( if the piezoceramic is subjected to an external electric field, it will be compressed or stretched depending on the value and direction of the electric field. ) with the reference ( ) or another reference.
11.Please provide the reference of the sentence ( electromechanical characterisation of these devices must be as reliable and accurate as possible such that~ ) with the reference ( Hwang, S. C., C. S. Lynch, and R. M. McMeeking. "Ferroelectric/ferroelastic interactions and a polarization switching model." Acta metallurgica et materialia 43.5 (1995): 2073-2084.) or another reference.
12.Please provide the reference of the sentence ( In previous works, the authors of this paper exposed some difficulties presented by the ANSI IEEE ) with the reference. It seems to be previous publications of the authors' work.
Author Response
Dear Ms./Mr.,
Several changes were made to the text and figures in response to the reviewers' "recommendations" and Editor' "recommendations". This letter provides a point-by-point response to the reviewers.
All changes in the revised manuscript were marked in red.
Kind regards
Francisco Javier Jiménez

Reviewer 2 Report
The authors do well comparisons ! The developed application evolution explained. LabVIEW a good choice. Maybe authors forget some references. The level of plagiarism of 8% indicate this.
Try to verify and add eventually forget references.
All the claimed quality of measurements and results are deep connected with the probe fixture. From the figure 4 this fixture do not looks well and we suggest to insert a short description and a better picture of the fixture.
I asked to add some idea about fixture of the samples - important in this kind of measurements.
From the pictures the readers can not understand if the fixture satisfy the reliability of the measurement.
Author Response

(The authors gave the same response as above.)
